# Australian general practice registrars perceived development of cultural safety for Indigenous patients: A qualitative study

Kay Brumpton[1,2,3]*, Hannah Woodall[1,2,3], Rebecca Evans[3], Henry Neill[3], Tarun Sen Gupta[3], Lawrie McArthur[4], Raelene Ward[5]

1 Griffith University School of Medicine and Dentistry, Rural Clinical School, Toowoomba, Australia, 2 Rural Medical Education Australia, Toowoomba, Australia, 3 James Cook University College of Medicine and Dentistry, Townsville, Australia, 4 The University of Adelaide, Adelaide, Australia, 5 University of Southern Queensland, Toowoomba, Australia

* k.brumpton@griffith.edu.au

## Abstract

### Objective

It is poorly understood how general practice registrars develop skills towards providing culturally safe care to Indigenous patients. Finding ways to enable the patient to determine culturally safe care, but not be responsible for teaching or correcting, is a challenge for health education systems. This study aims to explore how Australian general practice registrars perceive they develop towards cultural safety in readiness to consult with Aboriginal and Torres Strait Islander patients.

### Design

This study utilised a pragmatic philosophical position. It is a qualitative study involving administration of a survey for demographic details alongside semi-structured interviews to explore how general practice registrars develop towards cultural safety. Data collection was conducted from March to August 2022. Transcripts were studied using a content analysis approach.

### Setting

Regional Queensland, Australia.

### Participants

General practice registrars

### Results

Sixteen registrars completed the survey and semi-structured interview. Registrars described four main factors that contribute towards their development of cultural

**Data availability statement:** Data cannot be shared publicly because participants did not give consent for the data to be publicly available. As the number of Indigenous registrars enrolled with the GP training organisation is very small (approximately five), interview data could be potentially identifiable and attributed to these participants. Freely available data would breach compliance with the protocol approved by the Human Research Ethics Committee. Data are available from the James Cook University Human Ethics Committee (contact via ethics@jcu.edu.au) for researchers who meet the criteria for access to confidential data.

**Funding:** This project has received funding through an Australian College of Rural and Remote Medicine Education Research Grant. The funders had no role in study design, data collection and analysis, decision to publish, or preparation of the manuscript.

**Competing interests:** The authors have declared that no competing interests exist.

safety: shared or similar life experiences, learning from cultural safety training, experiential learning, and critical reflection. Registrars thought it was acceptable for patients to educate, teach or correct culturally unsafe care and did not reflect on the power imbalance in these relationships.

## Conclusion

Cultural safety appears to be a lifelong learning journey balancing critical self-reflection against an external determination of cultural safety. Understanding this process warrants further research and review of medical education teaching.

---

### Introduction

Australian Aboriginal and Torres Strait Islander people, hereafter referred to respectfully as Indigenous people, comprise just over three percent of the Australian population. The Australian Indigenous population is diverse with at least 167 traditional languages spoken in homes [1]. Australian health care professionals must acknowledge and respect this diversity and the health care needs of this population. The effects of colonisation have resulted in this population carrying a disease burden of 2.3 times that of non-Indigenous Australians [2]. Consequently, the Australian medical workforce must develop and deliver culturally safe care to improve access to healthcare for Indigenous people and address this health inequity [2].

Cultural safety is a key priority for improving the health care of Indigenous peoples. The teaching of cultural safety has been complicated by the wide variety of different terms and definitions in use [3]. However, in 2019, a consensus definition was released by the Australian Health Practitioner Regulation Agency (AHPRA) Aboriginal and Torres Strait Islander Health Strategy group. This definition was unique in having been derived in consultation with Aboriginal and Torres Strait Islander community. This definition now provides a consistent basis for teaching and learning cultural safety, as below:

> "Cultural safety is determined by Aboriginal and Torres Strait Islander individuals, families and communities. Culturally safe practice is ongoing critical reflection of health practitioner knowledge, skills, attitudes, practicing [sic] behaviors and power differentials in delivering safe, accessible and responsive healthcare free of racism" [4].

In Australia, primary care is largely provided through private general practices which ideally provide comprehensive, holistic, person-centred, and continuous care to patients [5]. The medical workforce in general practice consists of specialist general practitioners (GPs), non-specialist GPs and GPs in training (hereafter referred to as registrars). Registrars train to be specialised GPs under an apprenticeship model in both the hospital and community setting [6] and have core curricula requirements for demonstration of cultural safety [5,7].

The mechanisms whereby registrars develop cultural safety in Indigenous health are poorly understood. A 2016 integrative review of the literature on developing cultural competence in general practice reported on eight studies in Indigenous health [8]. None of these studies explored how GP registrars perceive they develop cultural safety or similar. Given the link between cultural safety and improving health of Indigenous Australians, it is important to understand the development of skills in this domain amongst learners who will become a considerable component of the workforce into the future.

This project seeks to explore how Queensland GP registrars perceive they develop towards cultural safety in readiness to consult with Indigenous patients.

## Methods

### Research design

We report our qualitative study according to the Consolidated Criteria for Reporting Qualitative Research [9]. A detailed description of the methods has been published elsewhere [10]. Briefly, this study is part of a larger study exploring assessment of GP registrars' cultural safety when consulting with Indigenous patients. This paper reports on a qualitative component exploring how GP registrars perceive they develop cultural safety to provide appropriate care to Indigenous patients.

The James Cook University Ethics Committee approved this study (H8296) following review by Aboriginal and Torres Strait Ethics Advisors in accordance with the National Health and Medical Research Council guidelines.

### Participants

All 562 GP registrars undertaking active training with a Queensland GP training organisation (GPTO), to obtain fellowship with The Royal Australian College of General Practice and/or the Australian College of Rural and Remote Medicine, were invited to participate in this study by email.

### Data collection

Participant recruitment and data collection was conducted during from 01 March to 31 August 2022. Registrar demographic details were collected by survey followed by video-conferenced semi-structured interviews with a research assistant to explore how GP registrars develop cultural safety. The research assistant had no prior relationship with the participants but was familiar with their training program. Participants provided both written and verbal informed consent. Written consent was provided when completing the survey and verbal consent recorded at the start of interviews.

### Data analysis

Authors had full access to all the deidentified data. Survey data was transcribed, and then descriptively analysed to characterise the participants who were also offered copies of the transcripts for member checking. Four main questions from the semi-structured interview were considered for this analysis:

How do you feel you developed cultural safety?
What has been the most effective way for you to develop this cultural safety?
What cultural safety (or similar) training have you completed?
What are the main things you have learnt from this training?

Transcripts were studied using a content analysis approach [11] using emerging data-driven codes. A1 initially coded the data. NVivo® analysis software and Excel were used when coding data, recording frequency of occurrence of item of interest, and collating key concepts. All authors reviewed the coding tree, associated data, and discussed construction of themes.

## Reflexivity

The principal investigator, A1, is an experienced GP academic working in an Aboriginal Medical Service, A2 a GP academic and public health registrar, A3 a senior health researcher, A4 an Aboriginal cultural educator for the GPTO, A5 academic GP, A6 GP academic and director of the GPTO, and A7 is an Aboriginal health academic. A community advisory group of Indigenous people have been involved in the research since inception.

## Results

### Participant characteristics

Sixteen registrars participated in this study. A further ten registrars indicated willingness to be interviewed but did not respond to a follow-up email to schedule an interview time. As data analysis suggested no new insights were being generated and thematic saturation had been reached after sixteen interviews, further attempts at contact were not made.

Overall, the registrar sample was comparable to the training cohort except with over-representation of junior doctors and under-representation of international graduates. Trainees from both Australian general practice training colleges were equally represented. Two participants identified as Aboriginal and Torres Strait Islander. Participant characteristics are detailed in Table 1.

Registrars described four main factors that contribute to their development of cultural safety (Fig 1).

**Table 1. Participant characteristics.**

| Registrar characteristic | | Participants n = 16 (%) | Cohort of registrars with the GP training organisation n = 562 (%) |
|---|---|---|---|
| Age | 25-34 years | 10 (62%) | 309 (55%) |
| | 35-44 years | 5 (31%) | 198 (35%) |
| | >44 years | 1 (6%) | 55 (10%) |
| Gender | Female | 11 (69%) | 327 (58%) |
| | Male | 5 (31%) | 235 (42%) |
| Post-graduate year | 1-4 years | 6 (38%) | 75 (18%)* |
| | 5-7 years | 8 (50%) | 224 (54%)* |
| | 8-10 years | 1 (6%) | 89 (21%)* |
| | 11-15 years | 1 (6%) | 30 (7%)* |
| | Not recorded | – | 144 |
| Experience in Aboriginal and Torres Strait Islander health | Nil | 8 | Not available |
| | <1 year | 4 | |
| | 1-3 years | 3 | |
| | >3 years | 1 | |
| Medical degree | Preferred not to state | 1 (6%) | 0 |
| | An Australian University in Queensland (the state where the research was conducted) | 11 (69%) | 295 (53%) |
| | Other Australian university | 4 (25%) | 120 (21%) |
| | International university | 0 (0%) | 147 (26%) |
| Time lived in Australia | 5-10 years | 5 | Not available |
| | 11-15 years | 1 | |
| | >15 years but not all their life | 2 | |
| | All their life | 8 | |

*n=418 as data not available for 144 registrars.

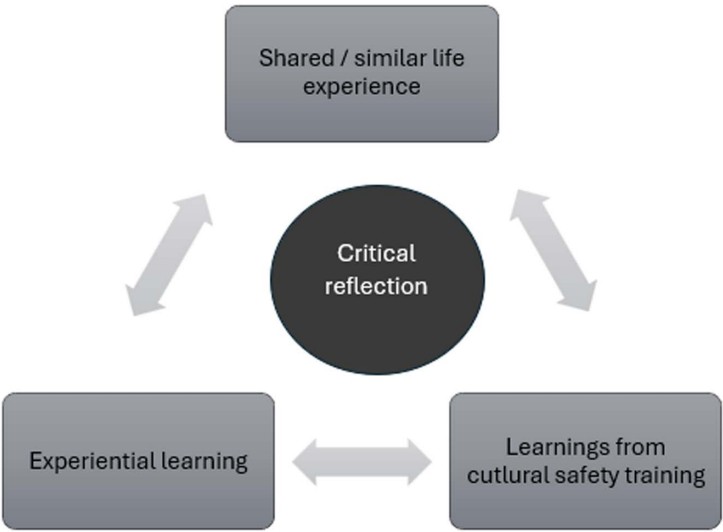

**Fig 1. Factors that contribute to the development of cultural safety.**

### Shared/similar life experience

Many registrars described the development of cultural safety through shared or similar experiences with Indigenous peoples (Table 2). These included both deficit and strengths-based experiences.

**1.Cultural safety (or similar) training.** All registrars had completed cultural safety training (or similar) as part of their university studies, hospital employment or GP training. These ranged from on-line modules, to 2–3-hour workshops to 1–2-day masterclasses with cultural immersions. Most registrars described the educational experience as positive, particularly if it was delivered with local context.

> *I found they were good if they were locally run and they were about my local area that I was working in or being in, but otherwise some of them it's not very transferrable.* [7400]

Notably, not all cultural training was a positive experience. One registrar described feeling distressed by formal cultural safety training. This distress was not related to grappling with the injustice of past wrongs or ongoing health inequalities. On one occasion, this registrar was upset by the unfairness of a simulated, constructed consultation not adequately reflecting his own experience with Indigenous people. Another time this registrar struggled with the tone of delivery of a workshop and not knowing how to process the emotion expressed by the speaker.

> *I went to a med school with a lot of people of different coloured skins. And we were doing sort of a cultural safety workshop in general practice, and it just felt like we were being yelled at the whole time... and we all sort of commented at the end, "Well, where do you go with that?" …. The overwhelming sense is that just sit there and shut up and wait for it to go through because if you say anything then you'll be labelled with the 'R' [Racist] word and which is like the death of you, like professionally.* (6389)

Learning the history of Indigenous people was the most frequently reported outcome of cultural safety training. Registrars also described learning about the epidemiology of disease, attitudes and particular practising behaviours and skills. Fig 2 presents a summary of registrar learnings, and these are detailed in Table 3.

**Table 2. Shared and similar life experiences.**

| Shared/similar factors contributing to the development of cultural safety | Example registrar quote |
|---|---|
| Heritage | … having lived as an Aboriginal [person], there is a cultural connection that I feel as though is portrayed with my patients when just in everyday communication and explaining things and having the Aboriginal family myself knowing would my mother understand this if I explained it in this particular way?...And it's just knowing to have that connection, to not assume that everybody has academic past or that they have health literacy. (8230) |
| Interests | I'm from [regional city/town]. I have a lot of shared interests outside of work in terms of sporting and outdoors and, possibly shared land. (9423) |
| Childhood experiences | I think also understanding troubled backgrounds as well. You know, I had my own troubled background when I was a kid. I lived in different care arrangements and stuff like that. (6389) |
| Minority group | And I think coming from a [non-Australian] background myself actually really helped in that sense because I know what it is like to have difference and be different and I guess made me able to appreciate and respect that more. (1131) |
| Physical appearance | This might sound a bit silly, but I think sometimes being darker skinned helps. I think sometimes they might feel a bit of a connection. (2601) |
| Experience of displacement | So, my family were refugees…so that's one of the reasons why I try and care about the experience of our Aboriginal Australians because I see a lot of similarities in terms of displacement. (4091) |
| Importance of family | When I have spoken to them [Aboriginal people], we also have a very strong cultural framework in which the families are very united. We also call our elders, uncle, and aunties, and they also call everyone uncle and aunties. So, when I have discussed, they have found that this is more closer to what their culture is. (1111) |
| Basis of religion | They [people] may find a similarity in what I believe. Most of them have changed the religion, but they still have their very core basic understanding of one monotheism or something like that. (1111) |
| Collective decision making | That a lot of their cultural practices are very similar to the [non-Australian] practices as well, where a lot of the things may be more a collective kind of thought rather than an individualistic opinion and decision. So, to me, I think I was able to kind of reflect back on my own culture and how it's quite similar, but not really sort of thing. (6434) |

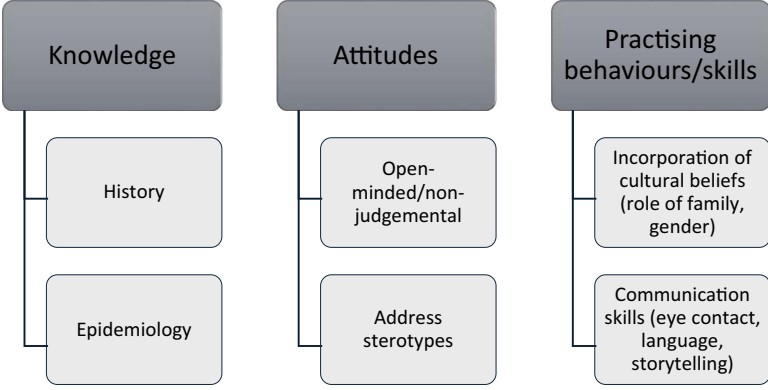

**Fig 2. Summary of registrars' main learnings from formal cultural safety training.**

**2. Experiential learning.** Most registrars described interacting with Indigenous people, and people from other cultures, as the most effective way to develop cultural safety.

*But I think it's just something that you kind of have to learn as you go. (1131)*

*And things start to become more natural when we talk about cultural safety and practice cultural safety. It goes from this sort of contrived, theoretical thing they sort of taught you in med schools…and then it slowly trickles and translates*

**Table 3. Learnings from cultural competency (or similar) training.**

| Theme | Sub-theme | Example registrar quote |
|---|---|---|
| Knowledge | Historical issues affecting Indigenous people | The thing that I took away from that [training] was understanding the history. So, it gave me a lot of context for why or how their culture had been affected and suppressed, or why people may be more hesitant to share it. And I think it just put everything in context of how if I am not somebody who can create a culturally safe environment, I will probably be perpetuating the issue that's already been ongoing for several years. (2601) What I have learnt is that from the way that the cultural safety and competency modules that I've completed have been put together, makes people really reflect on the history of Australia, what other people's experiences are, brings that to the sort of forefront that of their mind that this could be something that this particular person in front of me is experiencing and how can I make this transaction…culturally safe for this person? (8230) |
| | Epidemiology | So, I think firsthand experience learning about the culture, learning about the population and the health issues from an academic aspect during medical school and then as a health professional doing the job. (2797) |
| Critical reflection | Attitude | I think the main thing that I've learnt is that you just have to be open minded and not a d***head basically. (1131) So, there were some core principles that were spoken about just in terms of getting to know someone and being non-judgemental and using communication skills that were still transferable. (7400) |
| | Addressing stereotypes | …and that taught me a lot that a lot of Aboriginal people actually do occupy really high up…positions, and you won't always know that they're from an Aboriginal heritage unless you ask. (7216) We learnt about the sorry day, but how really that hasn't rectified the issue. And then we learned a little bit about the cultural background that has made not just the struggles that they've gone through, but the successes. So some of the Aboriginal Australians that are known for their art, their music, their sports and so the kind of things that we should focus on celebrating as well. (2601) |
| Practicing behaviours and skills – incorporation of cultural beliefs | Men's and women's business | And we were taken to where they [Aboriginal people] were living thousands of years ago and we were told from the very beginning what is their belief, how they actually are, sort of what was their interaction with the settlers and how it has sort of come to now in the modern day. What are their ideas with regards to gender health, female health and the role of the female and the male in the society? And how do we have to address these different sort of health concerns being GPs or being GPs in training? (1111) |
| | Role of family | The main thing is…like basically they [Aboriginal people] have their own culture and beliefs which I did not know much before the workshop. Like men's business, women's business and their beliefs regarding death and their beliefs regarding family and regarding everything. So, I think in a sense they help me to understand the way, for example, sometimes it's a grandmother presents with-, brings her granddaughter or grandson for review instead of their parents. (7358) |
| Practicing behaviours and skills – consultation and communication skills | Power differential | …it can be quite intimidating, having lots of standing people around the foot of your bed, especially if none of them look like you. (4091) |
| | Use of language and common slang | Understanding common slang terms and then appropriate words to use in return was another really important thing. (2601) |
| | Eye contact | So, like if there's no eye contact doesn't mean that the patient's not listening to you. It's part of the cultural aspect of it. (6278) So, I guess on a concrete level, communication skills would be one. And so, like some Aboriginal Australians especially, they prefer to avoid eye contact and find that quite an intimidating…as if you're exerting power over them. Whereas to a Western patient they'd find that as a sign of respect that the doctor was looking at them rather than looking at the chart. So that's something I've learnt and tried to practice. (4091) |
| | Storytelling | In providing patient education and just consultation in general, using a story to help deliver the information (4091) |

*into actual real-world practice, which also has to grow and develop alongside all the other skills. Like at one point when I put in a cannula as a med student, I was like shaking. And now I do it without even thinking. It's all these skills that unfold with you thinking about it less and less. (6389)*

Registrars described the value of learning from patients whilst on clinical placements as students and then further as junior doctors. Nearly half of the registrars described it as acceptable and appropriate for patients to educate, teach or correct culturally unsafe care.

*I think asking patients directly, "Is there anything that I say or do that has offended you or is there something in this process that might make you feel like you've been violated or harmed?". (3270)*

*Just asking the patients in particular, especially with Aboriginal and Torres Strait Islander patients, because I'm completely ignorant or I still am quite ignorant, but was even more so just when I ask things be saying, say, sort of be like, oh, "I'm not sure and I'm really sorry if this comes across as racist, but can I ask this or is it racist of me to ask this? Or is it racist for me to say this?" And just asking them if it's culturally appropriate, I guess, …or even asking them, is it culturally appropriate for me to be talking to you about this as a woman doctor, especially with men? Yeah. So, I think just asking because I think people are quite responsive to that and they're always more than happy to give feedback and teach. (1131)*

Some registrars were concerned about asking questions about culture and cultural safety out of fear of this being perceived as racist behaviour.

*And if they said something that is not consistent with the mainstream culture, I would just confirm with them. For example, like maybe they have different beliefs regarding death. I would just ask them, "Can you elaborate? Can you tell me more about your understanding, your beliefs about death, and what is the normal thing to do?" For example, if the mom is dying from a terminal illness, what's the correct culturally thing to do? (7358)*

*So, if they wanted to explore a concept that wasn't something that I'm familiar with, I would just ask the: "I don't know exactly what that means to you. Can you help me and teach me what it means and how you want me to take that further into the consultation?" (7216)*

Other registrars expressed a more proactive approach to providing culturally safe care in asking patients or Aboriginal Liaison Officers about their opinions and experiences.

*So, when I moved there, I really didn't know much about the culture but quickly picked up the ropes and a lot of it was learning from either the liaison officers or from the patients themselves and their family members. (6278)*

No registrars directly mentioned the leadership or expertise of cultural educators or cultural mentors. One registrar noted learning local history from their GP supervisor (1111). Another described learning from historical Australian drama films (3270).

**3. Critical reflection.** A registrar expressed the importance of ongoing critical reflection in the development of cultural safety:

*I think the most effective way has been experience, but then also reflection…on situations where it doesn't go right, where the patient does express they're unhappy or they haven't gone ahead with something that you wanted them to. And then working out the reasons why. Was because it was in conflict with some of their beliefs or their cultural practices? And so, then that led to I need to know a little bit more about this person, but also a little bit more about that culture or religion or ethnicity or something that's happened there (7400).*

An Indigenous registrar detailed their own experiences of feeling culturally unsafe in the work environment and how they drive non-Indigenous doctors to self-reflect.

*So, a lot of my [Indigenous] colleagues that have felt culturally unsafe, we have been adapting to how could we politely reword or ask for an explanation [from non-Indigenous doctors]? …So, in those circumstances I've said, "Oh, it's very interesting that you say that". When I had a colleague ask me what percent of Aboriginality I was, I said, "It's very interesting that you ask that and if I had given you any particular number, would that change your opinion of me? If I was to say 5%, 25%, 33% or 100%? What value do you give that number? So, I'm interested to ask you why you've asked*

*that particular question?" So, I think over time to develop cultural safety is to adapt to the environments and to learn to politely question why someone is asking you something that you may feel threatened, that you don't feel culturally safe in that environment. (8230)*

## Discussion

This study explored how GP registrars perceive that they developed towards cultural safety. Registrars described the development of cultural safety through shared or similar life experience, cultural safety training, experiential learning, and reflective practice.

By nature of being a qualified medical practitioner, registrars are in a position of relative privilege through both socio-economic and healthcare access. Registrars may not readily recognise this advantage and the implications of this position. Some registrars in this study had identifiers that simultaneously granted them both power and marginalisation within the Australian context [12]. Over half of the registrars in this study were born outside of Australia. As such, these registrars may not feel assimilated into the dominant culture. Furthermore, registrars work as junior doctors in a hierarchical structure and earn less than the average weekly ordinary time earnings for full-time adults [13]. These factors may influence the lens through which registrars view privilege and the social determinants of health and may pose a barrier to culturally safe care. Rather than pretending these sociocultural differences do not exist, Wilson et al [14] suggest reframing how these differences shape the patient-registrar consultation leading to improved outcomes. In contrast, registrars who feel they do not have shared commonality with Indigenous peoples, or recognise they benefit from social privilege, must not presume they are incapable of relating to Indigenous patients.

Registrars described key learnings of knowledge, attitude, practising behaviours and skills from formal cultural safety training. These learnings are commonly classified as cultural awareness training and has been the focus for most health professional training within Australia [15]. Registrars mentioned the importance of Aboriginal Liaison Officers in the development of cultural safety. Previous studies have also shown the benefit of cultural educators and cultural mentors in teaching about culture, history, and their impact on patients [16,17] however no registrars directly discussed their influence. This may reflect that registrars did not recognise cultural mentors or cultural educators who were delivering training. Alternatively, it may indicate that most registrars, particularly those not working within Aboriginal Medical Services, have limited access to cultural educators and cultural mentors. Universal access to cultural educators and cultural mentors may assist in improving development of cultural safety by GP registrars.

Most registrars described exposure to, and working with, Indigenous peoples as being fundamental to their development of cultural safety. Furthermore, registrars had an expectation that patients would be prepared to both teach cultural safety and correct any culturally unsafe behaviour observed amongst clinicians. Registrars did not always seem to engage in critical self-reflection to recognise the danger of culturally unsafe behaviour on patients (or Indigenous colleagues) or the effect of their own position of power within this relationship. Finding ways to enable the patient to determine culturally safe care, but not be responsible for teaching or correcting, is a challenge for health education system. We assert that it is inexcusable to expose patients to culturally unsafe care within the learning process. While people need to learn they should not do so in an unsafe way that may impact others. Patients have a right to feel safe; this study suggests we may be exposing patients to unsafe learners, who may put them at risk. Thus, a well-meaning educational approach – providing registrars with experience and exposure to Indigenous health – may in fact be detrimental to those it is trying to help. This 'hidden' problem may contribute to the health inequity gap and have wider impacts for training as well as accreditation and supervision.

Some registrars recognised the cultural burden and inherent risks of expecting patients to teach and remediate unsafe behaviours and instead chose not to ask questions. By defaulting to a place of their own safety rather than seeking out learnings from supervisors or other appropriate role models, registrars may have missed opportunities to develop cultural

safety skills. Similarly, the registrar with a negative experience of cultural safety training may have adopted "ad hoc coping behaviours" rather than adopting a self-reflective stance and focussing on how cultural identity is complex and unique to every individual [8 p7].

Registrars did not report learning that cultural safety must be determined by Indigenous peoples, or the importance of critical reflection. The registrars in this study had completed medical degrees from varying universities across Australia and have undertaken cultural safety training. Teaching and education on this fundamental basis of cultural safety appears not to influence registrars' perception of cultural safety. Making the invisible racism visible, whilst protecting patients from culturally unsafe care, requires an approach that does not tolerate any interpersonal or systemic racism [18]. Supporting registrars to understand their own cultural lens and examine the influence of their own culture on consultations may assist development and delivery of culturally safe care.

## Limitations

This is a small sample of GP registrars located in one Australian state with possible participation and acquiescence bias (i.e., registrars interested in cultural safety may be more likely to volunteer and give what they perceive as the right answer). If the registrars most interested in cultural safety have volunteered (and they have limited insight about their practice) – it is concerning to consider how others might be practising. In addition, our sample under-represented international medical graduates. Exploring how these registrars view cultural safety is an important area for further research.

## Conclusion

Cultural safety is a lifelong learning journey balancing critical self-reflection against an external determination of cultural safety. Understanding this process warrants further research and review of medical education teaching. There is need for further evaluations of cultural safety in GP training considering how cultural safety is both taught and assessed: who determines what is culturally safe care and what are the attributes of the GP registrar, as determined by the Indigenous community, that contribute to a culturally safe practitioner. Culturally safe health care will play an important role in redressing health inequities amongst Indigenous people. Understanding the development of cultural safety amongst learners is an important step towards supporting the development of a culturally safe primary health care workforce.

## Author contributions

**Conceptualization:** Kay Brumpton, Tarun Sen Gupta, Raelene Ward.

**Data curation:** Kay Brumpton.

**Formal analysis:** Kay Brumpton, Hannah Woodall, Rebecca Evans, Henry Neill, Tarun Sen Gupta, Lawrie McArthur, Raelene Ward.

**Funding acquisition:** Kay Brumpton, Lawrie McArthur.

**Methodology:** Kay Brumpton, Rebecca Evans, Henry Neill, Tarun Sen Gupta, Raelene Ward.

**Supervision:** Rebecca Evans, Tarun Sen Gupta, Raelene Ward.

**Writing – original draft:** Kay Brumpton.

**Writing – review & editing:** Hannah Woodall, Rebecca Evans, Henry Neill, Tarun Sen Gupta, Lawrie McArthur, Raelene Ward.

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
