## [Decision Letter · Decision Letter 0]

Dear Dr. Brumpton,

Please submit your revised manuscript by  Jan 10 2025 11:59PM. If you will need more time than this to complete your revisions, please reply to this message or contact the journal office at plosone@plos.org . A rebuttal letter that responds to each point raised by the academic editor and reviewer(s). You should upload this letter as a separate file labeled 'Response to Reviewers'.A marked-up copy of your manuscript that highlights changes made to the original version. You should upload this as a separate file labeled 'Revised Manuscript with Track Changes'.An unmarked version of your revised paper without tracked changes. You should upload this as a separate file labeled 'Manuscript'.

We look forward to receiving your revised manuscript.

Kind regards,

Oriana Rivera-Lozada de Bonilla

Academic Editor

PLOS ONE

3. Thank you for stating the following financial disclosure: This project has received funding through an Australian College of Rural and Remote Medicine Education Research Grant. 

4. In the online submission form, you indicated that the datasets are not publicly available due to participants being potentially identifiable from the small dataset but are available from the corresponding author on reasonable request.

Reviewers' comments:

Reviewer's Responses to Questions

**Comments to the Author**

1. Is the manuscript technically sound, and do the data support the conclusions?

Reviewer #1: Partly

Reviewer #2: Yes

Reviewer #3: Yes

2. Has the statistical analysis been performed appropriately and rigorously?

Reviewer #1: N/A

Reviewer #2: N/A

Reviewer #3: N/A

3. Have the authors made all data underlying the findings in their manuscript fully available?

Reviewer #1: Yes

Reviewer #2: Yes

Reviewer #3: Yes

4. Is the manuscript presented in an intelligible fashion and written in standard English?

Reviewer #1: Yes

Reviewer #2: Yes

Reviewer #3: Yes

Reviewer #1: 1. The introduction must give a global perspective to the discourse on cultural safety before presenting on the regional perspective, herein, Australian Aborginal and Torres Strait Islander people.

2. Present specific objectives or research questions that undergird the study in the last section of the introduction. Ensure that you elucidate why an academic probe into them is timely and relevant.

3. In the abstract it was indicated that 16 out of a total of 562 GP registrars were engaged in the study. You must detail this under the methods section, highlighting the inclusion and exclusion criteria for the recruitment of the study participants, the sampling design adopted and what informed the sample size.

4. There should be a separate section that explains how ethical protocols were duly followed in the conduct of the entire research process including the administration, and designing of instruments as well as in the conduct of the video conferenced interviews.

5. The survey instruments (survey and interview guide) must be explained. The scholarly procedures that were followed in the designing of the instruments, the survey questions, and scholarly procedures in validating them before their administration must be explained.

6. Enough scholarly justifications for the use of video conferenced interviews must be given vis-a-vis it's demerits when compared to face-to-face interviews. More importantly, how was objectivity maintained throughout the conduct of the interviews? These must be discussed.

7. Under the results, authors must interpret the participants' characteristics described and how they may have influenced the data garnered and their analyses.

8. No sound conclusions have been drawn and presented. The authors must draw sound and data-driven conclusions from the key results of the study presented. Also, recommendations for practice could be suggested to GP registrars based on the revelations, though humble, for the essence of cultural safety in medical delivery.

9. The manuscript needs to be thoroughly proofread to fix few identified syntax errors.

Reviewer #2: As noted in the manuscript, the low response rate resulted in a very small sample size. That said, the importance of this issue, and the insights gleaned from this small sample, render this worthy of publication with the hope that this paper will spark new conversations and, perhaps, follow-up studies with greater participation.

Reviewer #3: Abstract

The abstract should be restructured based on the following:

Background of the paper should describe the main objective or objectives of the study. You may include the study setting here briefly.

method section should briefly explain, which includes research design, study participants, data analysis, etc.

Results and findings should summarise the most important results; significance and conclusion should include implications. Refer to the journal guidelines https://journals.plos.org/plosone/s/submission-guidelines

Major comments

Background/objective: The study does not specify the location or settings where the registrars are practicing (e.g., rural, urban, or remote areas). This context is critical as it influences cultural interactions and healthcare challenges.

Design: The following elements are missing, e.g., the authors did not describe how participants were recruited, the sample size, or the inclusion and exclusion criteria. Briefly explain it was used in the study. The explanation of the design may not be necessary in the abstract.

Setting: delete and refer to previous comments

Participants: delete and refer to previous comment.

Results: Briefly state the study findings and implications

Conclusion: Include practical public implications.

Introduction

The introduction is succinct and good; however, the authors did not include an example of a health outcome or disparity (e.g., life expectancy, prevalence of specific diseases) to give readers a clearer picture of the issue. Also, the citations are referenced numerically (1, 2), but their specific relevance is not elaborated. It’s unclear where the data comes from or how authoritative it is.

Methods

Design

The authors did not explicitly explain the type of qualitative design used. While it mentions that it is a "qualitative component" of a larger study, it does not specify the exact qualitative approach (e.g., phenomenology, grounded theory, ethnography, case study, narrative analysis). This lack of clarity makes it difficult for readers to understand the framework guiding the research or how data was collected and analysed.

Participants

The number of participants in the study is not explicitly mentioned in the provided text. While 562 GP registrars were invited, it is unclear how many agreed to participate or were included in the final analysis.

Data collection

The authors did not describe how the survey and interview instruments were developed, including whether they were grounded in existing frameworks, literature, or stakeholder input (e.g., Indigenous advisors or GP registrars).

There is no mention of how the instruments were tailored to address cultural safety in a manner appropriate for Indigenous contexts.

The instruments' reliability and validity are not discussed. Without validation, there is a risk that the tools may not accurately capture the constructs of interest (e.g., cultural safety development).

The authors did not explain how the instruments (survey and interviews) were administered beyond the mode of delivery (e.g., videoconferencing).

Data analysis

The authors did not explain how rigour and trustworthiness were ensured during the coding and theme development process. While member checking and team discussions are mentioned, there is no detailed explanation of how biases were minimised, how inter-coder reliability was assessed, or how the "emerging data-driven codes" were systematically validated.

Results

The combination of tables and plain text in the presentation of the results makes it challenging for readers to follow. I recommend removing the tables and presenting the findings uniformly in plain text to enhance clarity and ensure easier comprehension.

I suggest the authors revise the presentation of the quotes, as they are not accurately or effectively formatted. E.g., I found they were good if they were locally run and they were about my local area that I was working in or being in, but otherwise some of them are not very transferrable. (7400). I also suggest the authors should use pseudonymization instead of numbers.

Discussion

The discussion is generally good; however, it lacks implications and suggestions for the phenomenon under investigation.

Limitations

While the limitation acknowledges the small sample size and geographic focus, it does not explicitly state how these factors affect the generalisability of the findings to other regions, populations, or healthcare settings.

While participation and acquiescence bias are mentioned, potential biases introduced during data collection or analysis (e.g., interviewer influence, coding bias) are not addressed.

Conclusion

The conclusion lacks specific, actionable recommendations for policymakers, educators, or healthcare organisations on implementing the findings in practice.

The conclusion does not acknowledge the study's limitations or how they might influence the interpretation of findings.

**Do you want your identity to be public for this peer review?** For information about this choice, including consent withdrawal, please see our Privacy Policy

Reviewer #1: **Yes: ** Dickson Adom

Reviewer #2: **Yes: ** Andrea J Alveshere

Reviewer #3: No

---

## [Author Response · Author response to Decision Letter 1]

30 Mar 2025

We have uploaded a document responding to the reviewer and editors comments (Response to PLOSOne reviewers_final).

---

## [Decision Letter · Decision Letter 1]

Australian general practice registrars perceived development of cultural safety for Indigenous patients: a qualitative study.

PONE-D-24-41188R1

Dear Dr. Kay Brumpton,

We’re pleased to inform you that your manuscript has been judged scientifically suitable for publication and will be formally accepted for publication once it meets all outstanding technical requirements.

Kind regards,

Oriana Rivera-Lozada de Bonilla

Academic Editor

PLOS ONE

**Comments to the Author**

Reviewer #4: (No Response)

2. Is the manuscript technically sound, and do the data support the conclusions?

Reviewer #4: Partly

3. Has the statistical analysis been performed appropriately and rigorously?

Reviewer #4: N/A

4. Have the authors made all data underlying the findings in their manuscript fully available?

Reviewer #4: (No Response)

5. Is the manuscript presented in an intelligible fashion and written in standard English?

Reviewer #4: (No Response)

Reviewer #4: (No Response)

**Do you want your identity to be public for this peer review?** For information about this choice, including consent withdrawal, please see our Privacy Policy

Reviewer #4: **Yes: ** Selorm Adinkra

---

## [Editor Report · Acceptance letter]

PONE-D-24-41188R1

PLOS ONE

Dear Dr. Brumpton,

I'm pleased to inform you that your manuscript has been deemed suitable for publication in PLOS ONE. Congratulations! Your manuscript is now being handed over to our production team.

Kind regards,

on behalf of

Dr. Oriana Rivera-Lozada de Bonilla

Academic Editor

PLOS ONE